# What Is or What Is Not a Risk Factor for Arterial Hypertension? Not Hamlet, but Medical Students Answer That Question

**DOI:** 10.3390/ijerph19138206

**Published:** 2022-07-05

**Authors:** Tomasz Sobierajski, Stanisław Surma, Monika Romańczyk, Krzysztof Łabuzek, Krzysztof J. Filipiak, Suzanne Oparil

**Affiliations:** 1Faculty of Applied Social Sciences and Resocialization, University of Warsaw, 00-927 Warsaw, Poland; tomasz.sobierajski@uw.edu.pl; 2Faculty of Medical Sciences in Katowice, Medical University of Silesia, 40-752 Katowice, Poland; monika.romanczyk@med.sum.edu.pl; 3Club of Young Hypertensiologists, Polish Society of Hypertension, 80-214 Gdańsk, Poland; 4Independent Researcher, 43-600 Jaworzno, Poland; labuzek@labuzek.com; 5Institute of Clinical Sciences, Maria Sklodowska-Curie Medical Academy in Warsaw, 00-136 Warsaw, Poland; krzysztof.filipiak@uczelniamedyczna.com.pl; 6Department of Medicine, School of Medicine, University of Alabama at Brimingham, Brimingham, AL ZRB 542, USA; soparil@uabmc.edu

**Keywords:** arterial hypertension, risk factors, medical students, knowledge

## Abstract

Hypertension is a leading cause of cardiovascular disease and premature death worldwide. The most important method of preventing hypertension is social awareness of its causes. An important role in educating society about hypertension is played by medical personnel. The study involved 327 students of medicine representing all years of study. The study used a proprietary questionnaire containing test questions about knowledge of the causes of hypertension (classical and non-classical factors), as well as questionable and false risk factors for the disease. The students’ knowledge of the complications of hypertension was also assessed. Most of the students rated their knowledge about hypertension as good. Classical risk factors for hypertension were identified by students in all years of study: I–III and IV–VI. Non-classical risk factors for hypertension were less often identified by the students. The students almost unanimously indicated that the complications of hypertension include heart failure, heart attack, stroke, aortic aneurysm, kidney failure, atherosclerosis, eye diseases and worse prognosis in COVID-19. Students’ knowledge of the causes of hypertension increased during medical studies. The knowledge of the respondents about classical risk factors for hypertension was extensive, whereas knowledge of non-classical risk factors it was insufficient. Most of the respondents were well aware of the complications of hypertension. Some students identified some factors incorrectly as increasing the risk of hypertension. Emphasis should be placed on the dissemination of knowledge about non-classical hypertension risk factors to medical students.

## 1. Introduction

Hypertension is a worldwide disorder. The study by Beaney et al. showed that about 34% of adults in the world suffer from hypertension [1]. Of this group, just over half (58.7%) were aware that they had hypertension and 54.7% were taking medication [1]. According to World Health Organization (WHO) estimates, 1.28 billion people between 30 and 79 years of age have hypertension, and only one in five people with hypertension has it under control [2]. The problem of hypertension, especially in middle-income countries, is so important that WHO developed a guide in 2021, presenting the most current evidence-based guidelines for the diagnosis and treatment of hypertension [3]. It should be noted that hypertension also increases the risk of severe COVID-19, as indicated by several studies and patient observations [4,5].

In Poland, the percentage of adult patients with hypertension, according to a study conducted on an unrepresentative group of volunteers by Sęk-Mastej et al., is 55.4% [6]. This study also found that 83% of hypertensive persons were aware that they had hypertension, whereas only 46.7% had controlled blood pressure (BP), that is <140/90 mmHg [6]. Hypertension is a risk factor for many diseases, including chronic kidney disease, aortic dissection, stroke and ischemic heart disease, significantly shortening life expectancy in the general population [7]. In Poland, hypertension, next to hypercholesterolaemia, is the most common risk factor for cardiovascular disease [8]. Furthermore, increased BP is a significant cause of death throughout the world [9]. According to the data of the Central Statistical Office in Poland published in 2016, cardiovascular disease was the cause of 46% of all deaths. The percentage of deaths caused by cardiovascular disease has decreased in recent years, but the situation remains troublesome [10].

The high incidence of hypertension is closely related to the high prevalence of risk factors for this disorder [11]. According to the classification adopted by WHO, risk factors can be classified as modifiable and non-modifiable [2]. Modifiable risk factors include those that are influenced by an individual’s attitudes and behaviors, such as excessive salt intake, smoking, low level of physical activity, being overweight, and excessive alcohol consumption. These are referred to as modifiable lifestyle factors (MLFs) [12]. Non-modifiable risk factors are those that are influenced by a person’s structural characteristics, rather than their behaviors. These include, but are not limited to, age, family history of hypertension, gender, and genetic makeup.

Some risk factors are generally recognized, whereas others are controversial. Those better recognized are referred to as classical risk factors (CRFs) in this study and those that remain controversial are labeled non-classical risk factors (NCRFs). CRFs include arterial hypertension in parents, low physical activity, obesity, high salt intake, smoking tobacco-containing cigarettes, high alcohol intake, increasing age and male gender [12,13,14,15,16,17,18,19,20,21]. NCRFs include daily smoking of electronic cigarettes, passive smoking, irregular sleep, obstructive sleep apnea, periodontitis, consumption of energy drinks, air pollution, environmental noise, use of painkillers and anti-inflammatory drugs, use of hormonal contraception and consumption of food products containing licorice [22,23,24,25,26,27,28,29,30,31,32,33]. There are also lifestyle and behavioral factors that are commonly perceived as risk factors for hypertension, but for which there is no clear evidence of an effect on BP (e.g., hiking in high mountains above 2500 m) [34,35]. In addition, there are factors that are generally regarded as increasing risk for hypertension, e.g., drinking 2–3 cups of coffee per day, for which we have evidence that they are not stimuli for hypertension [36]. For prevention of hypertension, it is important to know, especially for medical students—future physicians, about the lifestyle-dependent modifiable risk factors [37,38,39,40]. However, knowledge about the causes of hypertension in the general public is generally insufficient, and further education in this field is very important [41,42,43,44,45,46,47,48]. Since medical personnel play an important role in health education, it is important to teach medical students about the causes of hypertension.

The aim of this survey was to determine the state of knowledge among medical students about the causes of hypertension. Therefore, we posed several research questions. First, what is the level of knowledge about classical and non-classical risk factors for hypertension among students of each year? Second, how do students assess their knowledge of hypertension risk factors? Third, whether the knowledge of classical and non-classical risk factors of hypertension increases with the higher, clinical years of study? This study describes the level of knowledge of medical students about risk factors for hypertension, a critical health issue in the face of the escalating prevalence of hypertension worldwide.

## 2. Material and Methods

### 2.1. Design and Settings

The study used a quantitative survey method and was conducted online, using an original survey questionnaire, between November 2020 and March 2021. All medical students at the Medical University of Silesia in Katowice in Poland (ŚUM) were invited to participate in the study.

### 2.2. Sample Size

Of the 1800 medical students enrolled at the ŚUM at the time of the survey, 327 completed the questionnaire. The survey sample is representative of the total surveyed population of medical students at the ŚUM. With this sample size and the number of medical students at the ŚUM (*n* = 1800), the error margin was 3% (95% confidence level and proportion 50).

### 2.3. Questionnaire

A survey questionnaire was constructed specifically for this study. The questionnaire was based on the latest medical knowledge in the field of hypertension, analysis of the medical curriculum of medical universities, and knowledge of implementation of research in medicine. The questionnaire was based on our previous research experience in the study of medical students. It is divided into four sections. Section 1 includes 2 demographic questions relating to gender and year of study. Section 2 includes a question relating to the subjects’ self-assessed knowledge of hypertension that uses a numerical scale from 1 to 6, where 1 means “insufficient” and 6 means “excellent”. Section 3 lists 27 risk factors for hypertension. Section 4 lists 8 diseases that can result from hypertension. In the third and fourth sections of the questionnaire, a 4-point scale was assigned to each of the factors/diseases, a variation on the Likert scale where at one end of the scale is a “definitely yes” response and at the other end is a “definitely no” response. The “neither yes nor no” response was removed from the classic Likert scale. Instead, a “don’t know” response was added, which was off the scale. The questionnaire was piloted in 20 medical students to validate and evaluate the tool. Participants in the pilot study provided minor corrections that were incorporated into the final version of the questionnaire.

### 2.4. Data Collection

The survey was conducted online using the Google web platform. The link to the survey was made available to the survey coordinators, who distributed it to the ŚUM medical students. Participation in the survey was voluntary. The survey was confidential, and the survey coordinators ensured that it was not possible to link the participants with individual survey results.

### 2.5. Ethical Considerations

No personal information, including computer IP, was collected. Because of the anonymous nature of the survey and the inability to track sensitive personal information, the study did not require ethics committee approval.

### 2.6. Data Analysis

Quantitative and categorical variables were described with the methods of descriptive statistics. Cross-tabulations and the chi-squared test were used to evaluate the association of risk factors with students’ knowledge level. All statistical analyses were performed in IBM SPSS Statistics 27.0.1.0. For all analyses, a *p*-level of < 0.05 was considered statistically significant.

## 3. Results

### 3.1. Sample Characteristic and Self-Assessment of Knowledge

A total of 327 ŚUM medical students participated in the study. Two-thirds of the respondents *n*= 227 (69.4%) were females, who constitute about 70% of all medical students at ŚUM. First year students were the most responsive to requests to participate in the survey (*n* = 116, 35.5%) (Table 1).

Students rated their knowledge of hypertension on a grading scale ranging from a grade of 1, which is unsatisfactory, to a grade of 6, which is excellent. The highest number of students, one in three, rated their knowledge as good (*n* = 36.4%) and the lowest as excellent (*n* = 2.1%). On average, men rated their knowledge much better than women. Twice as many women as men rated their knowledge of hypertension with the lowest rating, and more than five times as many men as women gave the highest rating. The same percentage of women and men gave themselves a mediocre rating (Table 2).

Sixth-year students rated their knowledge of hypertension relatively high—nearly half of them (45.5%) gave themselves a very good grade. Very good and excellent grades were given by fifth-year students (34.2%), fourth-year students (25.0%), third-year students (16.7%), second-year students (14.9%), and first-year students (5.2%).

### 3.2. Assessment of the Influence of Risk Factors on the Incidence of Hypertension

Medical students at medical universities in Poland spend the first three years in pre-clinical classes, generally without patient contact, and the following three years in clinical classes, during which they have contact with patients. On this basis, it can be assumed that due to contact with patients and classes implemented in clinics, students of senior years (IV–VI) will have more knowledge about hypertension risk factors. Therefore, to assess the impact of risk factors on hypertension, we grouped the students by year. Group A includes students in years I–III (*n* = 199; 60.9%), and group B includes students in years IV–VI (*n* = 128; 39.1%).

Analysis of students’ knowledge of risk factors that influence hypertension revealed several statistically significant relationships (Table 3). A trend of increasing risk score for hypertension was observed for physical inactivity (*p* = 0.039), daily smoking of traditional cigarettes (*p* = 0.002), aging (*p* = 0.041) and being female (*p* = 0.005).

### 3.3. Evaluation of the Influence of Hypertension on the Occurrence of Diseases

The medical students surveyed concluded that hypertension may increase the occurrence of a variety of cardiovascular diseases (Table 4.). In particular, they found statistically significant relationships between hypertension and heart failure (*p* = 0.007) and stroke (*p* = 0.001). Interestingly, they did not find significant relationships between hypertension and myocardial infarction, kidney failure or atherosclerosis.

## 4. Discussion

Hypertension is the leading cause of cardiovascular disease and premature death worldwide [49]. Furthermore, the incidence of hypertension has doubled worldwide in the years 1990–2019 (648 million versus 1.278 billion) [50,51]. In Poland, 42.7% of adults have hypertension [52]. Elevated systolic and diastolic BP significantly increase the risk of death from any cause, cardiovascular events, coronary heart disease or stroke [53]. Thus, prevention of hypertension is an extremely important health issue. A review of the literature by Surma et al. found that the level of knowledge of Poles about the causes of cardiovascular diseases, including hypertension, was low [41,42]. Furthermore, knowledge about the causes of hypertension was low in both the general population and in patients with a diagnosis of hypertension [7,54]. People living in rural areas are particularly lacking in knowledge about the causes of hypertension. In one study, nearly 50% of people living in rural areas of India did not know a single risk factor for hypertension [45]. Older people also tend to have insufficient knowledge in this area. A study by Cielecka-Piontek et al. analyzed the knowledge about the causes of arterial hypertension in 154 elderly Poles and found that 63% did not know any or knew only one risk factor for hypertension [43].

An important element in the prevention of hypertension is population-wide awareness of the risk factors, particularly modifiable risk factors, for its development. The basis for medical intervention in the population is to recommend lifestyle modifications to minimize risk factors that both predispose to the development of hypertension and worsen BP control in people with existing hypertension [55]. Medical providers play an extremely important role in educating the public about the causes of hypertension and should be educated in this field early in their studies. In our study of medical students, later-year students assessed their knowledge of the causes of arterial hypertension better than earlier-year students (average 4.14 in IV-VI year versus 2.78 in I–III year). This may be because (as mentioned before) the first three years of medical studies in Poland are rich in basic science subjects, while the fourth year features clinical topics that include hypertension.

In our study, we assessed the knowledge of medical students (I–VI years) of classical and non-classical risk factors for hypertension. Moreover, we assessed their knowledge of other factors that are incorrectly classified by the public as increasing the risk of hypertension. It has long been known that hypertension in parents, sedentary lifestyle, obesity, excess salt in the diet, daily and occasional smoking of traditional cigarettes, heavy alcohol consumption, aging, and male gender are risk factors for hypertension [12,13,14,15,16,17,18,19,20,21]. In our study, we found that medical students, even in the early years of their studies, had a good understanding most of the classical risk factors for hypertension and that this understanding increased during their years of study. An exception was the occasional smoking of traditional cigarettes—nearly 35% of students said that this was not a factor that could increase BP.

Slightly different results were obtained by Shaikh et al. in a study of 110 students in the early years of medical studies. They found that the most common classical risk factors for hypertension were stress, obesity and smoking (75.5%, 77.6% and 71.8%, respectively). Risk factors such as excess salt in the diet, sedentary lifestyle, the presence of hypertension in parents, older age and male gender were significantly less frequently identified (69.1%, 47%, 50%, 40% and 12%, respectively). Knowledge of classical hypertension risk factors was lower in these students than in students in our study. The discrepancy between the results of our study and the study of Shaikh et al. may have resulted from differences in stages of training of respondents (our study included students in all years of medical study, whereas the study of Shaikh et al. included only students in the early initial years of medical study) [56]. The study of Zawadzki et al., carried out in 50 second-year medical students and published in 2007, also analyzed knowledge about the causes of hypertension. In this study, as in ours, 98% and 96% of students identified obesity and high salt intake as risk factors for hypertension, but a lower percentage of respondents identified hypertension in parents (75%), smoking (86%) and alcohol consumption (70%) as risk factors [57]. In our study, knowledge of the respondents about the influence of sedentary lifestyle and aging on the risk of hypertension increased significantly as their medical education progressed. These discrepancies indicate the need for comprehensive, evidence-based medical (EBM) education in this area beginning early in the medical school curriculum.

Students’ knowledge of non-classical hypertension risk factors, including smoking electronic cigarettes, passive smoking, irregular sleep, obstructive sleep apnea, periodontitis, consumption of energy drinks, air pollution, environmental noise, use of painkillers and anti-inflammatory drugs, use of hormonal contraception and consumption of food products containing licorice, was far less complete. These risk factors, although less well known, do have documented negative effects on BP [22,23,24,25,26,27,28,29,30,31,32,33]. Although students increased their knowledge of non-classical risk factors for hypertension in the course of their studies, they had, in general, a significantly lower level of knowledge of these than of the classical risk factors. For example, few upper-year students identified: irregular sleep (35%), periodontitis (>50%); environmental noise (>30%); use of non-steroidal anti-inflammatory drugs (nearly 50%); hormonal contraception (nearly 25%); and consumption of licorice (>30%) as risk factors for hypertension. In relation to non-classical versus classical risk factors for hypertension, the respondents more often answered “do not know”, indicating a lack of knowledge of the area.

A significant knowledge gap related to non-classical hypertension risk factors was also demonstrated by Shaikh et al. In their study, consumption of energy drinks and the use of oral contraceptives were identified by only 64.5% and 13.6% of participants as risk factors for hypertension [56]. Interestingly, in our study, knowledge about the harmful effects of smoking electronic cigarettes on BP decreased significantly over time (*p* = 0.002). Importantly, smoking electronic cigarettes increases BP and therefore can only be a tool in smoking cessation, not a substitute for traditional cigarettes [22].

In the case of factors whose influence on hypertension risk is uncertain, i.e., moderate alcohol consumption [19,58] and traveling to high mountains (over 2500 m above sea level) [34,35], opinions of the respondents were divided. In our study, 44% of students indicated that moderate alcohol consumption does not cause hypertension. Despite the uncertainly of the data in this regard, but taking into account that alcohol consumption is generally harmful to health (regardless of the amount) [59], there is a significant knowledge gap in the respondents. Importantly, the general population has similar misconceptions about the health effects of alcohol consumption. A study by Whitman et al. involving 5582 people showed that 1/3 believed that alcohol has a pro-health effect, whereas another 1/3 did not have any knowledge in this area [60]. Further research is needed to enhance knowledge of risk related to factors whose effects on health are now poorly understood.

Our study also assessed factors that are not regarded as increasing hypertension risk, e.g., having home pets, consuming 2–3 cups of black coffee daily, vitamin D deficiency and female gender [19,36,61,62]. Although a large percentage of respondents had no knowledge of these factors, almost 1/5 of respondents stated that having home pets increases the risk of hypertension, and nearly 60% expressed the incorrect belief that coffee increases BP. Similar results were reported in the studies by Zawadzki et al. and Shaikh et al. In these studies, coffee consumption was recognized as a risk factor for hypertension by nearly 64% and 35.5% of respondents, respectively [56,57]. In fact, a recent comprehensive review of the literature concluded that regular daily consumption of 2–3 cups of black coffee may reduce the risk of hypertension [36].

Similarly, over 40% of university students believed that vitamin D deficiency leads to the development of hypertension, although such a relationship has not been demonstrated in clinical studies [62]. Interestingly, over a quarter of the respondents indicated that female gender predisposes to the development of hypertension, and over 90% of senior medical students considered pregnancy to be a risk factor for hypertension. The physiological state of pregnancy cannot be classified as a risk factor [63].

Our results indicate that most students were well aware of most of the classical risk factors for hypertension but had a much lower level of knowledge about the non-classical risk factors. Importantly, a large percentage of students incorrectly identified some factors as risk factors for hypertension (factors that are incorrectly classified in the social awareness as increasing the risk of hypertension). The level of students‘ knowledge of the causes of hypertension was generally insufficient, in the authors’ opinion. The study by Shaikh et al. also found that the knowledge of first-year medical students about modifiable and non-modifiable risk factors for hypertension was insufficient [56]. Similar conclusions were reached by Rehman et al. in a study of the knowledge about hypertension in medical students and young doctors in Pakistan [64]. Furthermore, a study by Ślusarska et al., which included final-year medical students, found that their knowledge of cardiovascular prevention strategies was insufficient [65]. Zawadzki et al. also concluded that the knowledge of a medical student is only slightly greater than that of an average Pole [57].

Taking into account the enormous scale of the problem of hypertension in the population, knowledge about its causes should be widely disseminated, even in the early years of medical studies. Only a high level of education in the medical population, including medical students, about the causes of hypertension can increase public awareness and reduce the global burden of the disease. Owing to the importance of hypertension in public health and the evident lack of medical knowledge in the field, special emphasis should be placed on educating medical students about hypertension. An excellent approach would be to create permanent thematic blocks devoted to hypertension in the medical curriculum. This contrasts with the current situation, in which hypertension is often the subject of optional classes in which it is not possible for all students to participate.

The current study has some important limitations. First, it was carried out among students from only one medical university in Poland. Although the study sample included a relatively large number of participants. the results should be only cautiously extrapolated to medical students elsewhere in Poland and in other countries. Nevertheless, it may serve as a helpful pilot for research on this topic at other medical universities, as well as a starting point for discussion on the optimal form of hypertension education for medical students in general. Second, as our study shows that undergraduate students’ knowledge about non-classical risk factors for hypertension is insufficient, further in-depth studies should be conducted to find out the causes. For this reason, in future studies, attention should be paid to maintaining the proportion of students in the study concerning the year of study so that the oldest students are adequately represented. Third, the study may have omitted some risk factors for hypertension Although the authors made a great effort to ask about both the so-called classical and non-classical risk factors, other, non-specific risk factors for hypertension, are frequently overlooked.

## 5. Conclusions

Data obtained from this study indicate that medical students have good knowledge of classical risk factors for hypertension, but insufficient knowledge of non-classical risk factors for this very prevalent condition and have misclassified some risk factors. For this reason, and because of the enormity of the hypertension problem in both developed and undeveloped societies, it is appropriate to pay greater attention to hypertension risk factors in medical education. In particular, it is important adapt the teaching of hypertension risk factors to the cultural and social changes that underlie the enormous increases in prevalence of hypertension and resultant cardiovascular diseases that are now taking placed throughout the world.

## Figures and Tables

**Table 1 ijerph-19-08206-t001:** Study group characteristics and self-reported knowledge of hypertension (*n* = 327).

		*n*	%	Average Self-Esteem of Knowledge about Hypertension
Sex	women	227	69.4%	
	men	100	30.6%	
Year of study	I	116	35.5%	2.78
	II	47	14.4%	3.28
	III	36	11.0%	3.71
	IV	68	20.8%	4.06
	V	38	11.6%	4.08
	VI	22	6.7%	4.14
	for all			3.47

**Table 2 ijerph-19-08206-t002:** Self-assessment of students’ knowledge about hypertension: distribution of grades (*n* = 327).

	Grade		*n*	%
Self-assessment	unsatisfactory	all	17	5.2%
		female	14	6.2%
		male	3	3.0%
	mediocre	all	57	17.4%
		female	40	17.6%
		male	17	17.0%
	satisfactory	all	75	22.9%
		female	60	26.4%
		male	15	20.0%
	good	all	119	36.4%
		female	87	38.3%
		male	32	32.0%
	very good	all	52	15.9%
		female	25	11.0%
		male	27	27.0%
	excellent	all	7	2.1%
		female	1	0.4%
		male	6	6.0%

**Table 3 ijerph-19-08206-t003:** Assessing the impact of potential risk factors that cause hypertension (*n* = 327).

Factors	Year of Study A (*n* = 199) (B *n* = 128)	Definitely Not or Rather Not		Definitely Yes or Rather Yes		Do Not Know		*p*-Value
		*n*	%	*n*	%	*n*	%	
Hypertension in parents	A	9	4.5%	183	92.0%	7	3.5%	0.658
	B	4	3.2%	124	96.9%	0	0.0%	
Lack of physical activity, sedentary lifestyle	A	4	2.0%	195	98.0%	0	0.0%	0.039
	B	1	0.8%	127	99.3%	0	0.0%	
Obesity	A	1	0.5%	198	99.5%	0	0.0%	0.120
	B	0	0.0%	128	100%	0	0.0%	
Excess dietary salt	A	8	4.0%	188	95.5%	3	1.5%	0.402
	B	2	1.6%	126	98.4%	0	0.0%	
Daily smoking of traditional cigarettes	A	5	2.5%	190	94.5%	4	2.0%	0.443
	B	1	0.8%	126	98.4%	1	0.8%	
Daily smoking of electronic cigarettes	A	23	11.6%	163	81.9%	13	6.5%	0.002
	B	7	5.5%	104	81.3%	17	1.3%	
Occasional smoking of traditional cigarettes (e.g., once a month)	A	85	42.7%	105	52.8%	9	4.5%	0.140
	B	39	30.4%	84	65.7%	5	3.9%	
Passive smoking (second-hand smoking)	A	26	13.1%	167	83.9%	6	3.0%	0.253
	B	11	10.1%	110	86.0%	5	3.9%	
Occasional alcohol consumption	A	114	57.3%	66	38.1%	9	4.5%	0.271
	B	56	43.8%	67	52.4%	5	3.9%	
Frequent alcohol consumption	A	10	5.0%	175	93.0%	4	2.0%	0.123
	B	4	3.2%	123	95.3%	2	1.6%	
Having home pets	A	136	68.4%	20	9.0%	43	21.6%	0.122
	B	87	68.0%	23	18.0%	18	14.1%	
Working in shifts (irregular sleep)	A	80	30.2%	94	47.3%	25	12.6%	0.439
	B	23	17.9%	96	65.0%	9	7.1%	
Obstructive sleep apnea	A	110	55.3%	49	24.6%	40	20.1%	0.665
	B	17	13.3%	105	82.0%	6	4.7%	
Daily consumption of 2–3 cups of black coffee	A	62	31.2%	127	63.8%	10	5.0%	0.182
	B	50	49.0%	63	57.0%	5	3.9%	
Periodontitis	A	63	31.6%	77	38.7%	59	29.6%	0.625
	B	32	32.8%	53	41.4%	33	25.8%	
Vitamin D deficiency	A	86	43.3%	61	20.6%	52	26.1%	0.781
	B	45	35.2%	55	42.9%	28	21.9%	
Consumption of energy drinks	A	23	11.5%	170	85.5%	6	3.0%	0.800
	B	13	10.1%	132	87.5%	3	2.4%	
Aging of the body	A	18	9.0%	178	89.4%	3	1.5%	0.041
	B	9	7.0%	118	92.2%	1	0.8%	
Female gender	A	129	64.8%	31	15.6%	39	19.6%	0.050
	B	82	64.1%	34	26.6%	12	9.4%	
Male gender	A	68	34.2%	100	50,2%	31	15.6%	0.670
	B	22	17.2%	96	75.0%	10	7.8%	
Air pollution (smog)	A	55	27.6%	128	64.4%	16	8.0%	0.290
	B	24	18.7%	95	84.2%	9	7.0%	
Environmental noise	A	82	41.2%	91	45.8%	26	13.1%	0.861
	B	32	25.0%	88	68.7%	8	6.3%	
Traveling to high mountains (above 2500 m)	A	77	38.7%	84	42.2%	38	19.1%	0.754
	B	41	32.1%	70	54.7%	17	13.3%	
Use of painkillers, anti-inflammatory drugs	A	78	39.2%	82	41.3%	39	19.6%	0.118
	B	51	39.9%	65	50.8%	12	9.4%	
Use of hormonal contraception	A	62	31.1%	105	52.8%	32	16.1%	0.115
	B	28	21.9%	93	72.7%	7	5.5%	
Pregnancy	A	59	29.6%	111	55.8%	29	14.6%	0.446
	B	9	7.1%	117	91.4%	2	1.6%	
Eating foods containing licorice	A	72	35.2%	59	29.7%	68	34.2%	0.070
	B	21	16.4%	87	68.0%	20	15.6%	

**Table 4 ijerph-19-08206-t004:** Effect of arterial hypertension on the occurrence of diseases (*n* = 327).

Q: Can Arterial Hypertension Lead To?	Year of Study A *n*= 199 B *n* = 128	Definitely Not or Rather Not		Definitely Yes or Rather Yes		Do Not Know		*p*-Value
		*n*	%	*n*	%	*n*	%	
Heart failure	A	3	1.5%	195	98%	7	0.5%	0.007
	B	0	3.2%	122	100%	0	0%	
Myocardial infarction	A	7	3.5%	189	95%	3	1.5%	0.280
	B	0	0.8%	128	100%	0	0%	
Stroke	A	15	8%	181	90%	1	1%	0.001
	B	0	0%	128	100%	0	0%	
Aortic aneurysm	A	13	6.5%	179	90%	7	3.5%	0.186
	B	2	1.6%	126	98.4%	0	0%	
Chronic kidney disease	A	30	15.1%	151	75.9%	18	9%	0.669
	B	1	0.8%	127	99.2%	0	0%	
Atherosclerosis	A	45	22.6%	144	72.4%	10	5%	0.078
	B	7	5.5%	104	81.3%	17	1.3%	
Eye (retina) diseases	A	28	14.2%	157	78.8%	14	7%	0.171
	B	39	30.4%	84	65.7%	5	3.9%	
Death in COVID-19	A	26	13%	150	75.4%	23	11.6%	0.579
	B	10	7.8%	107	83.6%	11	8.6%	

## Data Availability

The data presented in this study are available on request from the corresponding author.

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
