# Peer review of "What Is or What Is Not a Risk Factor for Arterial Hypertension? Not Hamlet, but Medical Students Answer That Question"

_ijerph, 2022, doi:10.3390/ijerph19138206_

Round 1
Reviewer 1 Report
The manuscript investigates the level of awareness among medical students of various grades about risk factors for hypertension, and the findings are very interesting and meaningful.
This study makes us realize that even among medical students, knowledge of non-classical risk factors for hypertension is still quite lacking. This suggests that among the general public who lack professional medical knowledge, knowledge about the risk factors of hypertension is even more lacking, and it also indicates that the prevention and control of hypertension is still facing greater pressure.
The article adopts a cross-sectional survey, the overall design is simple but scientific, the research results are relatively detailed, and the discussion is also relatively sufficient. But I think it is slightly less representative in the sample. In the included sample, there are fewer senior students, and the authors should discuss the possible bias.
In addition, we should note that knowledge of non-classical hypertension risk factors is still insufficient among senior medical students. For this phenomenon, it is recommended that the author further search for the reasons (such as not covered in the textbook?) and point out ways to further improve this phenomenon.
Author Response
Reviewer I
The manuscript investigates the level of awareness among medical students of various grades about risk factors for hypertension, and the findings are very interesting and meaningful.
This study makes us realize that even among medical students, knowledge of non-classical risk factors for hypertension is still quite lacking. This suggests that among the general public who lack professional medical knowledge, knowledge about the risk factors of hypertension is even more lacking, and it also indicates that the prevention and control of hypertension is still facing greater pressure.
The article adopts a cross-sectional survey, the overall design is simple but scientific, the research results are relatively detailed, and the discussion is also relatively sufficient. But I think it is slightly less representative in the sample. In the included sample, there are fewer senior students, and the authors should discuss the possible bias.
In addition, we should note that knowledge of non-classical hypertension risk factors is still insufficient among senior medical students. For this phenomenon, it is recommended that the author further search for the reasons (such as not covered in the textbook?) and point out ways to further improve this phenomenon.
Thank you very much for your excellent evaluation of our research paper and the article prepared based on it.
We also thank you for your very insightful analysis and for pointing out the places that needed improvement. We have considered them all.
We have included a comment regarding the underestimation of subsequent year students in the study and added a paragraph in Limitations, line 333-337:
“Since, as our study shows, undergraduate students' knowledge about non-classical risk factors for hypertension is insufficient, further in-depth studies should be conducted to find out the causes. For this reason, in future studies, attention should be paid to maintaining the proportion of students in the study concerning the year of study so that the oldest students are adequately represented.”
Reviewer 2 Report
General Comment:
- The instruction of each section must be removed.
Introduction:
- It is well described the significance of hypertension. However, it's lack of the background and significance related to the aim of the study. The authors may focus on and the literature should support why the determination of the state of knowledge among medical students about the cause of hypertension is important and needed. How does the study contribute to the field?
- What would be your research questions or the hypothesis that the study would like to test or investigate?
Methods:
- The questionnaire was based on authors' previous research experience. How you ensure the questionnaire is valid and reliable in assessment the medical students' knowledge. Having a reference on the questionnaire how it was developed and constructed is highly recommended.
- It's unclear how these questions are constructed. What the rationales on selecting each factors.
Results:
- Why the self-assessment knowledge of the students were categorized by gender, not the year of school like other assessment i.e., I-III vs VI-VI.
- The results are too short and lack of sufficient detail. What actually means about the statistically significant relationship. Line 184: The relationship about hypertension and factors? I think it's not correct. The author divided into two groups of students. So you would like to see whether the knowledge of the I-III year students and IV-VI year students are significant or not.
- IV-VI students are able to get more correct answers that I-III students.
Discussion:
- The discussion focused on overall students' knowledge; however, the results were presented in a comparison between I-III students' and IV-VI student's knowledge on hypertension factors. I think this is may be because the study doesn't have a sound purpose of the study.
- "The aim of this survey was to determine the state of knowledge among medical students about the causes of hypertensions. This study describes the level of knowledge of medical students about risk factors for hypertension" - The results didn't report state of knowledge of students whether excellent, good, insufficient (Only use one self-report assessment). In addition, the results didn't summarize in term of knowledge. It just reported the questions that are statistical significant. Those questions did not test the state of knowledge. It compares the knowledge between two groups of students.
Overall
I think the study did not explicitly provide the convincing background and significance why the study is in need and can contribute to the field. The questionnaire doesn't have in-depth information to assess medical students' knowledge. Yes/no question are quite difficult to capture their knowledge and understanding especially among the medical students. It's unclear on the scales and criteria to determine the state of knowledge of these students. The implication of this study finding is unclear.
Author Response
Reviewer II
Thank you very much for your evaluation of our research paper and the article prepared based on it.
Introduction
General Comment:
- The instruction of each section must be removed.
We are very sorry, but unexpectedly there were technical requirements for the article in the manuscript, which have been removed.
Introduction:
- It is well described the significance of hypertension. However, it's lack of the background and significance related to the aim of the study. The authors may focus on and the literature should support why the determination of the state of knowledge among medical students about the cause of hypertension is important and needed. How does the study contribute to the field?
To what purpose our research can serve we refer in Conculsions.
- What would be your research questions or the hypothesis that the study would like to test or investigate?
Thank you for this comment. We have added a paragraph with sophisticated research questions and what the field can gain from this research. Lines:102-106
“Therefore, we posed several research questions. First, what is the level of knowledge about classical and non-classical risk factors for hypertension among students of each year? Second, how do students assess their knowledge of hypertension risk factors? Third, whether the knowledge of classical and non-classical risk factors of hypertension increases with the next year of study?”
Methods:
- The questionnaire was based on authors' previous research experience. How you ensure the questionnaire is valid and reliable in assessment the medical students' knowledge. Having a reference on the questionnaire how it was developed and constructed is highly recommended.
- It's unclear how these questions are constructed. What the rationales on selecting each factors.
For complete reliability of the sociological study, the questionnaire was subjected to a pilot study, as mentioned in the line: 162-164. Such evaluation/validation of the tool is a necessary part of the methodological process to check whether the questions in the questionnaire are constructed correctly, whether the questionnaire has a good design, whether the phrases used in the questions are understandable, and whether the scales used to assess knowledge are understandable.
Since there are many risk factors for hypertension, we decided to select a few dozen. For the classic risk factors, we decided to select those that are most prevalent. In the case of non-classical risk factors, however, we based our selection on a review of the literature and knowledge and research experience of the authors, who are hypertensiologists.
Results:
- Why the self-assessment knowledge of the students were categorized by gender, not the year of school like other assessment i.e., I-III vs VI-VI.
Students' self-assessed knowledge was categorized by both gender and year of study, including each year of study - Table 1.
- The results are too short and lack of sufficient detail.
According to the principle of Evidence-Based Medicine recommended by all major medical journals and scientific societies, the results found in the tables are not described in the text, hence perhaps the impression that the results are not well described. On the other hand, the results presented in the tables and the description underneath them give a very detailed description of the study results.
- What actually means about the statistically significant relationship.
We pay attention to statistical significance in our study, because, statistical significance is an assessment of whether an outcome occurs by chance. When an outcome is statistically significant, it is unlikely to occur by chance due to random fluctuations. There is a cutoff point used in determining statistical significance. A difference is called statistically significant if the occurrence of available data (or even more extreme data) would be unlikely, assuming the difference is absent. This expression does not mean that the difference should be large, important, or significant in the general sense of the word.
- Line 184: The relationship about hypertension and factors? I think it's not correct.
Thank you for that comment. The sentence is about the relationship between knowledge of factors and prevalence of hypertension. We have corrected that. Line: 213
- The author divided into two groups of students. So you would like to see whether the knowledge of the I-III year students and IV-VI year students are significant or not.
The decision to divide the students into two groups from I-III and IV-VI was based on the nature of student education at medical universities in Poland, in terms of pre-clinical and clinical classes. This is described in line: 178-183.
As this may not be fully understood, we have added line 207-209:
“On this basis, it can be assumed that due to contact with patients and classes implemented in clinics, students of senior year (IV-VI) will have more knowledge about hypertension risk factors.”
- IV-VI students are able to get more correct answers that I-III students.
This is what we wanted to test in the study. And as the results of our study show, it is not obvious.
Discussion:
- The discussion focused on overall students' knowledge; however, the results were presented in a comparison between I-III students' and IV-VI student's knowledge on hypertension factors. I think this is may be because the study doesn't have a sound purpose of the study.
The purpose of the study was clearly stated in the Introduction, line: 101-102
In that type of scientific article, the purpose of the discussion was to capture the results of the conducted study in the general context of research on hypertension risk factors with consideration of the recent literature. The process of medical education around the world is diverse, and indeed the purpose of the discussion is not to replicate the results that have been discussed above in Results. Nevertheless, in the discussion, we repeatedly refer to the breakdown we used in the study (lines: 218-223, 224-227, 229-233, 241-245, 260-268).
- "The aim of this survey was to determine the state of knowledge among medical students about the causes of hypertensions. This study describes the level of knowledge of medical students about risk factors for hypertension" - The results didn't report state of knowledge of students whether excellent, good, insufficient (Only use one self-report assessment). In addition, the results didn't summarize in term of knowledge. It just reported the questions that are statistical significant. Those questions did not test the state of knowledge. It compares the knowledge between two groups of students.
Using a Likert scale, we can determine the level of knowledge about a given risk factor for hypertension and whether the student has any knowledge. For example, suppose a student answers that air pollution "definitely does not" or "rather does not" impacts the occurrence of hypertension. In that case, we can state with a high probability that their knowledge of this risk factor is very low or low.
Results are given for all risk factor questions that were used in the questionnaire (Table 3.). We did not use a censoring or restriction method because this would not be in line with the research methodology. The results given in Results are not only the results of questions that are statistically significant, but of all questions.
Overall
- I think the study did not explicitly provide the convincing background and significance why the study is in need and can contribute to the field.
This was described in Conclusions.
- The questionnaire doesn't have in-depth information to assess medical students' knowledge.
Our study presented here is quantitative. In social research methodology, this study answers the question: “What is?”. Therefore, the quantitative research results for people looking for an answer to the answer: “Why is it so?” may feel disappointed. However, for the search for the answer: “Why is it so?” qualitative research serves. To be able to ask, however: “Why is it so?” one must first answer the question of what we are dealing with. Moreover, we answered this question in the context of students' knowledge of risk factors for hypertension. Thanks to this study, which we write about in the discussion, we can not only plan the process of educating students but also construct a qualitative study that will allow us to investigate in-depth the reasons for students' level of knowledge.
- Yes/no question are quite difficult to capture their knowledge and understanding especially among the medical students. It's unclear on the scales and criteria to determine the state of knowledge of these students. The implication of this study finding is unclear.
In formulating the scale for the questions, we used one of the best known and researched attitude and knowledge level scales, the Likert scale, as we write about in Questionnaire 2.3. At the same time, we note that we did not use a dichotomous yes/no scale in the study, we used an expanded scale that examined the degrees of knowledge.
*Moderate English changes languages and style are required
One of the authors of our article is Professor Oparil, who is native American, Section Chief of Vascular and Hypertension at the University of Alabama-Birmingham, who checked the article in great detail not only for content but also for language.
Reviewer 3 Report
The article of Sobierajski et. all “What is or what is not a risk factor for arterial hypertension? Not Hamlet, but medical students answer that question” is an interesting article about the level of knowledge among medical students about the causes of hypertensions. This is very important topic especially keeping on mind the high incidence of hypertension in the human population. It is very interesting that medical students have good knowledge of classical risk factors for the hypertension, but insufficient knowledge of non-classical risk factors.
Just a few minor comments, hoping they will be helpful.
Page 1 rows 38-43 should be deleted.
Page 2 rows 98-101 should be deleted.
Page 3 rows 102-112 should be deleted.
Page 7 row 220 change VI-VI to IV-VI.
Author Response
Reviewer III
The article of Sobierajski et. all “What is or what is not a risk factor for arterial hypertension? Not Hamlet, but medical students answer that question” is an interesting article about the level of knowledge among medical students about the causes of hypertensions. This is very important topic especially keeping on mind the high incidence of hypertension in the human population. It is very interesting that medical students have good knowledge of classical risk factors for the hypertension, but insufficient knowledge of non-classical risk factors.
Just a few minor comments, hoping they will be helpful.
Page 1 rows 38-43 should be deleted.
Page 2 rows 98-101 should be deleted.
Page 3 rows 102-112 should be deleted.
Page 7 row 220 change VI-VI to IV-VI.
Thank you very much for your excellent evaluation of our research paper and the article prepared based on it.
We also thank you for your very insightful analysis and for pointing out the places that needed improvement. We have considered them all.
We are very sorry, but totally unexpectedly there were technical requirements for the article in the manuscript, which have been removed.